# Perception, Self, and Zen: On Iris Murdoch and the Taming of Simone Weil

**Silvia Caprioglio Panizza** [ID]

Centre for Ethics, Department of Philosophy, Faculty of Arts and Philosophy, University of Pardubice,
532 10 Pardubice, Czech Republic; silvia.capriogliopanizza@upce.cz

**Abstract:** How do we see the world aright? This question is central to Iris Murdoch's philosophy as well as to that of her great source of inspiration, Simone Weil. For both of them, not only our action, but the very quality of our being depends on the ability to see things as they are, where vision is both a metaphor for immediate understanding and a literal expression of the requirement to train our perception so as to get rid of illusions. For both, too, the method to achieve this goal is attention. For both, finally, attention requires a dethronement of the self, considered as the source of illusion. In this paper I investigate what moral perception means for each of these philosophers and how it operates through attention and its relationship with the self. I will show that, despite many striking similarities, Murdoch's project does not equal 'Weil minus God', but offers a different concept of the self, a different understanding of its removal, and therefore a different picture of attention and moral perception. In evaluating both views, I will gesture towards a third way represented by Zen Buddhism, which both philosophers variously consider but do not embrace.

**Keywords:** moral perception; attention; self; Iris Murdoch; Simone Weil; Zen Buddhism

## 1. Moral Perception in Iris Murdoch

The question of perception preoccupied both Iris Murdoch and her great source of philosophical inspiration, Simone Weil. In an early essay, Murdoch writes that it is 'perfectly obvious' that 'goodness is connected with knowledge . . . a refined and honest perception of what is really the case' ('The Idea of Perfection' 330)[1] [1], and in *Metaphysics as a Guide to Morals*[2] [2], she claims to be offering an 'argument from perception' when considering the good quality of consciousness that is closer to perceiving the world as it is (MGM 254) [2]. In MGM, Murdoch returns to discussions of perception in Plato, Hume, Merleau-Ponty, Husserl, Wittgenstein, and others. The moral relevance of perception, for Murdoch, can be understood in two interrelated ways. First, it lies in the fact that perception reveals something about the perceiver's 'quality of consciousness'—a concept she does not define, but describes, for instance in 'Vision and Choice' [3][3], as how people think, what they think, and their 'total vision of life, as shown in their mode of speech or silence, their choice of words, their assessment of others . . . what they find funny' (VCM 80–1) [3]. Individual consciousness both shapes and colours perception, and also directs it towards and away from specific objects, so that not only how, but *what* one sees depends on one's moral quality of consciousness ('we see different worlds', VCM 82) [3]. Since consciousness is inherently evaluative for Murdoch, in her view, all perception is moral perception. As Cora Diamond [4] wrote, according to Murdoch, 'we are perpetually moralists'.

This thought on its own could lead us to read Murdoch as suggesting that we construe moral reality based on our inescapably evaluative consciousness, and indeed, there are places, especially in the earlier work, where she seems to make just this suggestion.[4] In turn, this thought could entail that reality itself is morally inert, and the mind projects value onto it, or frames a non-moral reality, according to value categories. To avoid this conclusion, which would lead us very far indeed from Murdoch's thinking, we need to

consider the other way in which perception, for Murdoch, is morally relevant: that is, the perception of a moral reality, where value is discovered in the world through both intellect and senses: ' . . . the word *discovery* is very much in place here. One is just not *inventing* it out of oneself, one is *finding it out* . . . ' [5] (p. 110).

How do we 'find out' value in the world? Murdoch's answer: through attention. The exercise of attention, for Murdoch, is how we can put our consciousness to good use. Furthermore, Murdoch tells us 'a good quality of consciousness' consists in 'the continual discrimination of true and false' (MGM 250) [2]. Attention is a truth-directed faculty, in whose very exercise we become better,[5] and it is the faculty that allows us to 'see things clearly'. Now, it is possible and tempting to take Murdoch's frequent use of visual terms only as metaphors, especially if the concept of metaphor is understood narrowly, and reject the idea that she is talking about moral perception proper. While it is true, as we shall see presently, that Murdoch's moral perception does not quite fit contemporary models, first, her talk of moral vision is not just an elaborate way to refer to a direct form of moral understanding, and second, Murdoch's idea of metaphor is not of something 'added on' to what is real, but rather for her metaphor is itself part of the way we understand, and indeed, perceive reality (see e.g., SGC 363–4) [6]. Several commentators have offered accounts of Murdochian moral perception (see [7–10]). One of the earlier and most sustained discussions of moral perception in Murdoch is Lawrence Blum's [11], which argues that moral perception is required both for the perception of a situation *as moral* (and hence, also for the application of theories which themselves do not rely on, or support moral perception), and for the perception of what is *salient* in a situation, which in turns allows us to construe the situation as something that can be correctly described in thick moral terms e.g., as racist, unjust, kind, selfish, careless, etc.

Despite attempts to put Murdoch in conversation with contemporary discussions on moral perception, Murdoch's conception does not fully fit into any of them. One reason, defended by Evgenia Mylonaki [9], is that Murdoch's idea of moral perception finds its main application in *attending to individuals*, where is the central point is not the perception of specific moral properties of the individuals, regarded as independent of the sensibilities of the perceiver (what Mylonaki calls the 'object-view', being the dominant one in contemporary discussions). Although there is currently an alternative view of moral perception, following McDowell [12], which takes moral perception to be the perception of reason-giving properties (what Mylonaki calls the 'reasons-view'),[6] a view that can be more easily accommodated in Murdoch's philosophy, Mylonaki reminds us that Murdoch's perception of the individual goes further than what is reason-giving, to include cases of moral perception where no appeal to our practical concerns is required (such as the M&D case) [9] (pp. 585–588). A third reason (as I defended in [8], is that Murdoch's conception of consciousness and moral reality is such that there is no distinction between natural and moral properties, and therefore, even the appeal to supervenience that characterises the more optimistic defences of moral perception, including Murdochian ones, is based on a distinction that she rejects.[7] Attention, paraphrasing Murdoch, is what allows us to see a flower as well as another person's needs (cf. OGG 357) [13]. There is no fundamental difference either in the faculty, or in the object, of such perceptions.

## 2. Unselfing, or Removing the Bad Self

Murdoch's premise for her idea of moral perception through attention is that, normally, we do not perceive the world aright, and that the main obstacle is the self. As she writes in 'On God and Good'[8] [13], 'all *just* vision . . . is a moral matter . . . the same virtue (love) [is] required throughout and fantasy (self) can prevent us from seeing a blade of grass just as it can prevent us from seeing another person' (OGG 357) [13]. All perception is moral perception. Attention can make that perception both clearer and morally good; however, in order to do so, it needs to overcome the self. What is so wrong with the self?

In the earlier essays in *Sovereignty*, Murdoch appeals to a Freudian view of the self as a 'mechanism' (OGG 341) [13], whose main goal is to protect itself from harm and

suffering, and to that end, it construes elaborate *fantasies* that distort reality representing it as we wish (or fear). Therefore, to be able to perceive the world as it is, the 'enemy' to be overcome is nothing short of the self.[9] Murdoch coins the term 'unselfing' to indicate this path. Many questions have been raised regarding the potential paradoxical nature of such an enterprise: How can *we* become morally better by overcoming ourselves? Who is the self that becomes better if the self is removed? Where do we find the energy and resources if 'we' are supposed to be nothing?

The main strategy to address these worries has been, I believe correctly, to point out that although Murdoch does say that we need to silence and overcome the self and to 'silence and expel self' (OGG 352) [13], if we read the idea of 'unselfing' in the context of her philosophy, what it really refers to is just one aspect of tendency of the self, namely that of distorting reality. This distortion of reality can be called variously selfishness, self-centredness, and self-absorption; thus, it is not the self as such that must go, but the bad (or, with Plato, the 'lower') part of it; the self is not removed, but reconfigured. This is the line taken in different ways by commentators whose readings are otherwise diverse, such as Antonaccio [14], Mole [15], Fredriksson and Panizza [16], Hämäläinen [17], and Meszaros [18].

There are philosophical and scholarly reasons to take this view. The latter appeal to Murdoch's insistence on the recovery of the individual and her 'quality of consciousness' as essential elements of a good and realistic moral philosophy.[10] The philosophical reasons appeal, as above, to the difficulty of accounting for moral perception if there is no 'self' who perceives, especially if moral perception is taken to be, as Murdoch suggests, at the same time, reality-driven and personal, or an individual apprehension of an independently given reality, where experience, concept-mastery, and good character traits all play a role in perceiving what is there. Some defenses of moral perception have employed precisely this argument by suggesting that, far from being an impediment, the knowledge and virtues of the self are required for perception to be accurate, and to perceive what we otherwise would not perceive.[11]

Read this way, the role of attention in revealing reality through unselfing presents a familiar picture of moral activity, one that Murdoch justly stresses as deserving more consideration than it typically receives in ethical theory: moral activity begins not with deliberation and judgment, but with perception; the kind of people we are influences our perceptions; we can be evaluated morally based not only on what we do, but on what we think and *see*. It follows that the first and foremost task is working on our perception, making it clearer, and removing impediments coming from a self that, while in some ways is helpful, is also an obstacle.

Murdoch's criterion for moral perception, then, is mostly a negative one. We do not have fixed standards to know whether we are perceiving clearly, but we know that perception that is not filtered by the self's intrusion is better. Such perception is a moral achievement in itself, regardless of the object, because overcoming the self is a moral achievement. Attention is ipso facto a form of unselfing, because when we attend we are turned away from the self. Attention is a 'direction of consciousness'. Depriving the self of its power involves turning our consciousness—our energies—to something else, anything else, as long as it is truly attended to.

In this understanding of attention, we can see just how different Murdoch's concept of attention is from the one employed in much of the psychological literature, where attention tends to be a morally neutral faculty of selection.[12] The key difference lies in the fact that while, for Murdoch, attention is indeed a direction of consciousness through the senses to an external object, such direction also implies a specific quality of consciousness: to attend, to direct consciousness to something external—and even, but not necessarily, to sustain this direction—is at the same time to perform two other acts: to desire, however implicitly, to know the object and, in parallel, to detach oneself from all that interferes with such desire to know, which includes primarily one's desires and pursuits, in short, one's self. Another

way of naming the desire implicit in attention is *eros*: love for the truth (cf. OGG 357; MGM 494–6) [2,13].

### 3. The Self in Attention: A Tame and a Radical View

Taking the familiar concept of attention to its further consequences, Murdoch shows how attention allows us to *see things as they are*. Lack of attention prevents us from hearing a rude remark in a conversation as well as a chorus in a song. The main reasons why attention is absent or fails stem from the concerns of the self. Conversely, when we attend, all that occupies our consciousness is the object of attention. If, as Murdoch suggests, attention gives us an inkling of how things are *regardless of us* (regardless of the particular self who is attending), then the concept of unselfing begins to take a more radical meaning. In attention, the object occupies our consciousness completely; there is no question of anything else, no question of an 'I'.

If we push this thought further still, we can reach the idea of attention as an ideal end point in which the object is so fully present that the perceiver becomes a mere vehicle for the manifestation of the object. The substantial self becomes transparent. Not a part of the self, but the self as such has to give way in the full realization of attention. This different interpretation of the self in attention has implications beyond the self: for the meaning and activity of attention, and for the understanding of perception as enabled by attention, which, in this view, becomes not a personal and active engagement with reality, but a passive allowing of reality to manifest itself. The former view of the self, as substantial and part of the historical individual, allows us to retain a familiar and commonsense understanding of who we are, our continuity over time, and the necessity of the individual point of view; here, attention is the activity of the individual, and successful moral perception consists of clearer perception made possible by knowledge, experience, and virtues.[13] The latter view of the self takes seriously the idea that attention reveals a reality that is there regardless of us. The ideal of reaching that reality implies a removal of the self that is only possible, in metaphysics, where the self is, indeed, something that can be removed, or something that never existed in the first place. Following a distinction introduced in [19], I call these the 'tame' and 'radical' views, respectively. These labels aim to signal that, on the former view, we do not need to give up a familiar and well-motivated concept of self even in the context of attention and moral perception; while, on the latter, the removal of the self targets not only parts or configurations of the self, but the self as such, thus, leading to a view of moral perception that is wholly impersonal, and, in which, value arises not from the interaction of self and world, but from the world only.[14]

In what follows, I will present the second, 'radical' view, through the work of Simone Weil. By presenting Weil's concept of attention and the role of the self, I will also highlight significant differences between Weil and Murdoch. In most accounts of Murdochian attention, the concept is just presented as derived from Weil, taking Murdoch's own word, and without further questioning (see e.g., [14] (p. 140), [20–23], and my past self included in [8]). While these accounts do not *claim* that the concept is the same, they allow for the impression that it is, an impression which I believe has built up in Murdochian scholarship. The key difference, when observed, is that Murdoch talks about the Good as a secular object and an animating force of attention, while Weil talks about God. Justin Broackes, for instance, commenting on Murdoch's radio talk on Weil in 1951, acknowledges that Murdoch needed to make Weil her own by modifying her ideas, especially removing 'the requirement of a God' and including 'a new affirmation of the independence (and, if properly understood, freedom) of human persons' [24] (p. 19); meanwhile, he reaffirms the idea that Murdoch's inheritance of Weil is primarily that of maintaining similar concepts without God: 'A great achievement of Murdoch's is to see how the methods can be fundamentally the same in the domain of morality as Weil had sketched for the domain of religion' [24] (p. 19). What I will suggest is that, if the idea of attention is different enough, the methods are not *fundamentally* the same. Since Murdoch, as I have argued, proposes a substantial view of the self in

attention, the danger is that Weil's much more radical proposal will be 'tamed' if read mainly as a religious version of Murdoch's secular concepts.

There are several reasons that make this misreading understandable. One is that Murdoch explicitly appeals to Weil as a source of inspiration for her key concepts, e.g., here:

> I have used the word 'attention', which I borrow from Simone Weil, to express the idea of a just and loving gaze directed upon an individual reality. I believe this to be the characteristic and proper mark of the active moral agent. (IP 327) [1]

Second, while Murdoch refers to Weil, she often does so in rather vague terms, without explaining to what extent she draws on her and where she departs, for instance here:

> Let me now simply suggest ways in which I take the prevalent and popular picture [of the moral subject] to be unrealistic. In doing this my debt to Simone Weil will become evident. (OGG 340) [13]

Finally, a more comprehensive reason for the conflation is that while Murdoch offers a different view of attention and the self from Weil, she is tempted by Weil's proposal, but her commitment to 'the individual' prevents her from following Weil down the path of attention and moral perception as impersonality. I suggest that Weil's influence remains for Murdoch both a tempting but untreadable path, as well as unfinished business. We can see this open-endedness, intriguingly, in the fact that she almost ends MGM [2] (in the penultimate chapter) with reflections on the 'void', a key Weilian concept, and raises more concrete worries about suffering, affliction and hunger—as if the Weilian spirit came back to remind her that on these fundamental questions ethics alone will always be embarrassed.

### 4. Attention as Decreation

When Murdoch picked up the concept of attention from Weil, she was drawing on something central to Weil's thought. One of the main collections of Weil's writings is meaningfully entitled *Attente de Dieu*[15] and in another collection, put together by her friend Gustave Thibon, Weil writes:

> The authentic and pure values—truth, beauty and goodness—in the activity of a human being are the result of one and the same act, a certain application of the full attention to the object. (*Gravity and Grace* 120)[16] [25]

Putting together attention to God and the idea that true values result from attention, we have a picture of Weil's extensive concept of attention: both ontological and ethical, a duty that is at once towards other individual beings and towards being itself. We attend to the world as God's creation, as the evidence of God's absence-as-presence, and as the object of God's love[17]. Hence, the world makes a demand on us to see it correctly.[18] Seeing the world correctly means attending to it.

As in Murdoch, for Weil, attention is a form of *erotic* desire for reality. That is a different kind of desire from any other. Ordinary desire is paradoxical and destructive, its natural aim is to consume and possess its object, thus turning it into something else and therefore destroying its own object. True desire—the desire of attention—is aimed at something which is absent, and therefore impossible to possess: god, or the good: 'God who is none other than the good itself—the good which is found nowhere in this world' (GG 94) [25]. Hence, 'because to desire something is impossible, we have to desire what is nothing' (GG 95) [25]. For Weil, desire for the real through god/the good is the only true animating desire, and it is what animates and sustains attention. Based on what we have seen above, Weil's influence on Murdoch is clear and, so far, their thoughts adhere quite closely.

Further, for Weil, as for Murdoch, attention goes hand in hand with a removal of the self. Weil knows that we must, as Murdoch tells us, not distort the reality that we are contemplating with our wishes, fears, and fantasies. She also knows, as does Murdoch, that this is almost impossible to do. But here also is where the two philosophers depart, with, I will argue, significant consequences for the meaning of attention, moral perception, and the general nature of the moral task.

For Murdoch, attention enables moral perception, in the dual sense of a morally laden perception of a moral reality, by combining the individual's engagement with the world through her characteristics and faculties, most importantly the *imagination*, with the removal of the self insofar as it distorts reality through self-protective *fantasy*. With the concepts of imagination and fantasy, Murdoch distinguishes between the good (realistic) and the bad (distorting) contributions of the self. 'Unselfing', then, refers to undoing the latter, not the former.

For Weil, the self has no room in attention. If attention is supposed to reveal reality as it is, then that reality has to be perceived without any influence or filter, including the perceiver herself. True perception, for Weil, is this: 'To see a landscape as it is when I am not there' (GG 42) [25]. In exceptional moments of attention, we are struck by a reality that is indeed other, and which fills our consciousness so fully that there is room for nothing else. To perceive rightly, we have to withdraw. That is precisely the imitation of God's self-withdrawal that Weil calls for: if it was God's withdrawal that created us, then to love God (i.e., wanting God to fully be), we have to renounce this creation. If attention is desire for God or good, in attention, we become nothing. This is 'decreation' (*décréation*): to give up the self that has been given us, to let the whole of reality be. Attention is not just selfless, but de-created; not a removal of selfish preoccupation, but a deconstruction of what makes us these particular selves. The imagination, so dear to Murdoch, is for Weil, always an expression of a self that manipulates reality, anxiously 'fill[ing] up empty spaces' (GG 48) [25]. It is not just the reflexive self or the self as an object that is the problem. The very existence of a point of view, though empirically inescapable, is an interference. This is attention for Weil: 'In such a work all that I call "I" has to be passive. Attention alone—that attention which is so full that the I disappears—is required of me' (GG 118) [25]. This is how attention and decreation together work towards realism: 'The only way to truth is through one's own annihilation' ('Human Personality' [HP], 27) [26].

This notion of the self is of course puzzling and paradoxical, hence, its radical nature; yet Weil does not shirk from its difficulties. If the self is nothing, who is performing the important task of attending? Contradictions abound. There must be someone who is the subject of these actions, these perceptions. Weil's claim is that there is no 'must'. Things are exactly as they seem to be in our moments of most extreme attentive rapture. There is only what we see. These thoughts may be best understood by a mystical sensibility, such as we find, for instance, in Anne Carson's [27] comparison of Weil, Margarete Porete, and Sappho, where the mystic, for Porete, swims in a sea of joy, where 'she feels no joy for she herself is joy' (p. 196). The similarity with Weil is striking: 'Perfect joy excludes even the very feeling of joy, for in the soul filled by the object no corner is left for saying "I"' (GG 31) [25].

The pressing ethical question, in this paradox, is not who attends but how do we fulfil our duties to others, if we are nothing. As Yoon Sook Cha [28] writes, 'One *has* to give (self renunciation being the archetype of this donation), but the question becomes what one exactly *has* in the first place to give', answering: 'in effecting one's "disappearance" one does not, however, relinquish one's obligation to the other. . . . Instead, the difficulty lies in being bound to the other through this very renunciation of one's "I"' (p. 3). The renunciation allows us to see something which is there independently of us. For Weil, chief among these ethical 'facts' is the fact that every living being cries out not be harmed, and resisting the self means resisting the tendency to give way to the mechanical law of 'force' that the self would wield on the other.[19]

The capacity of attention to unself on the one hand, and to decreate on the other, consists, in both cases, of a *direction* of consciousness and energy to something external but which we cannot and will not possess, with an attitude of love and detachment at the same time.[20] This may give the impression that Murdoch and Weil are achieving, after all, the same end in practice: turn away from your own concerns by filling yourself with reality, and you will see more clearly. But here, too, we must be careful to distinguish the significance of this move between the two cases. In Murdochian unselfing, the direction of attention away from the self 'purifies', in her words, a self that tends toward illusions, making us

better moral individuals. In Weilian decreation, the direction of attention without desire (or with objectless desire) is what presents the real as everything—and reveals the self to be, in fact, nothing. That is possible only if the self is made up of reflexivity, an illusion that we carry forward by taking ourselves to be substantial, but vanishes as soon as we look away. As Weil writes: 'the self is only the shadow which sin and error cast by stopping the light of God, and I take this shadow for a being' (GG 40) [25]. For Murdoch, the self creates illusions, including illusions about its importance; for Weil, the self *is* an illusion.[21]

Here we have come to what I suggest is the fundamental difference between Murdoch and Weil, based on two different conceptions of the self, which in turn separate more than usually acknowledged, their notions of attention and perception (as well as others that are not the concern of this paper). Yet, there are tendencies in Murdochian attention, starting from her language of unselfing and its suggestion of a more radical removal of self, and including her attraction to mysticism as an elements of the ethical, which take her closer to Weil, but only at times, and introduce ambivalences which explain the difficulty of specifying her inheritance of Weil. One of the more intriguing expressions of such ambivalence is Murdoch's discussion, in MGM [2], of 'pure consciousness', and her association of Weil with Zen Buddhism. Zen here emerges as an obvious comparison for our purposes too: the school of thought where attention is an absolutely central practice, and where the self is undone but in a way that makes no ontological commitments. In what follows, then, I propose Zen as a third way for the self in attention, one that both Murdoch and Weil appreciate but neither endorse.

## 5. Murdoch's Ambivalence and the Zen Way

Buddhism, Masao Abe writes, is 'quite unique in the history of human thought' precisely because it denies 'the existence of an enduring and unchanging soul or self' [29] (p. 68). Out of this illusory concept, the Buddha believed, all the evil in the world can be derived. What exactly is meant by no-self in Buddhism is the object of much debate, and varies depending on which branch one considers. Here I will focus on Zen Buddhism, for the dual reasons that it is in its discussion that Murdoch expresses clearly her dissatisfaction with the wholesale rejection of the self in attention while associating Zen with Simone Weil, and it provides the clearest alternative to both Murdoch's and Weil's concepts of the self, while retaining crucial similarities.

Murdoch was clearly attracted to Buddhism, to the point of calling herself a 'Buddhist Christian' (MGM 419) [2]. This remark, and her various mentions of Buddhism in her philosophy and representations in the novels, have intrigued readers [30,31]. There is much in Buddhism that appeals to her project, as Beran and Marchal [32] have remarked: 'the zazen; meditation techniques as exemplified in mysticism without worshipping any deities; the compassionate and realistic 'metaphysics' directed towards the present; occasionally a bit of picturesque ritual and magic' (186). However, as they write, 'Hardly does any actual school of Buddhism properly impersonate all these traits' (186), and Murdoch's her discussions of Buddhism tend to be vague, often invoked as an alternative to Christianity which avoids rituals and personalised objects of devotion. Nonetheless, she was not blind to the similarities between some Buddhist ideas and her interest in non-illusory perception, attention, and self-purification. While in her mentions of Buddhism, particularly in the novels, Murdoch often seems to be implying an understanding of Theravāda Buddhism [32] (p. 185), in MGM [2] her most sustained discussion concerns Zen Buddhism, to which she refers through her reading of Katsuki Sekida's *Zen Training* [33]. Here, Murdoch presents Zen as an alternative to Husserlian phenomenological reduction, and quotes Sekida on meditation as a discipline for 'root[ing] out the emotionally and intellectually habituated mode of consciousnes' and 'suspending every involvement of the personal ego' (Sekida in MGM 240) [2].

The mention of these Zen ideas need point no further than Murdoch's own idea of unselfing as removing the ego's interference that comes from habit and self-gratification. But she knows Zen goes further than that, and she begins to follow Sekida down that path:

'In pure cognition there is no subjectivity and no objectivity. Think of the moment your hand touches the cup: there is only the touch' (Sekida in MGM 243) [2]. Murdoch recognises that not only does Zen meditation root out the selfish ego, it also dissolves the dualism of subject and object. If the dualism is removed, so is the self. Murdoch glosses: 'No self, no subject, observes the serene waters of the lake' (MGM 243) [2]. She acknowledges that the idea is not without attraction, and perhaps even that the practice of attention may suggest this kind of conclusion:

> The notion of achieving a pure cognitive state where the object is not disturbed by the subjective ego, but where subject and object simply exist as one is here made comprehensible through a certain experience of art and nature. (Dualism is overcome: not such an arcane idea after all.) A discipline of meditation wherein the mind is alert but emptied of self enables this form of awareness, and the disciplined practice of various skills may promote a similar unselfing, or 'décréation' to use Simone Weil's vocabulary. Attend 'without thinking about'. (MGM 254) [2]

Here, Murdoch is putting her finger on something of extreme importance: the idea that in Zen meditative practice, dualism is overcome, and the self is neither an instrument of perception, nor a problematic entity that we need to destroy. The 'emptiness' (śūnyatā) that constitutes the alternative to existence and non-existence is called, in Mahayana Buddhism, the 'Middle Way', as Abe [34] reminds us, precisely because it is 'neither an eternalist view which insists on the existence of an unchanging, eternal entity as the ultimate, nor an annihilationist view which maintains that everything is null and void' (59). This is why, in Zen teachings, the masters do not reply to the question, urgently posed, 'what is the self?'. Both affirming and denying would reinstate the existence of something. For this reason, in both Dōgen and Takuan, we observe the *forgetfulness* of the I. This is also why Zen uses the phrase 'not-two' to signal its overcoming of dualism: it bypasses the standard dichotomy of self-other without affirming a monistic metaphysics (Nagatomo [35] pp. 213–44).

The focus on experience here, especially the experience of attention as exercised in meditation, is what leads to the forgetfulness of the self, and the overcoming of the dualistic standpoint. This method brings a simplicity that is attractive, as Murdoch sees. However, she cannot take this step. Her idea of attention comes with a perception that is purified, but only of self-reflexivity and possessiveness. The idea of removing the self is, indeed, too radical, as she writes: 'There is an "extremism" in what Sekida is suggesting, he advocates a lengthy ascesis at the end of which some purer and better state of consciousness and being is achieved' (MGM 243) [2].

Here, we need to return to the quote above from MGM, and note the introduction of Simone Weil in the discussion. Weil is mentioned in the same breath as Sekida as someone who went too far, who practiced an 'extremism' that is characterised by Murdoch as the loss of what is good about the self: the images, the personal imagination, the world we love precisely because we are separate. We may feel Murdoch's sense of a loss of warmth in this description.[22]

Weil and Zen are brought together in these pages of MGM as both 'extreme' and 'severe', indicating a criticism that is as philosophical as it is moral. Attention reveals the world as it is, but for Murdoch, it cannot reveal a world without us. Separate from us, yes, but not devoid of us. A few pages later Murdoch quotes Weil's remarks on Zen in her notebooks: 'The idea behind Zen Buddhism: to perceive purely, without any admixture of reverie (my idea when I was seventeen),' and comments:

> The imageless austerity of Zen is impressive and attractive. It represents to us 'the real thing', what it is like to be stripped of the ego, and how difficult this is. (Plato's distance from the sun.) Simone Weil felt a natural affinity with this 'extremism' which indeed she practised in her own life. She had studied Hindu and Buddhist philosophy. (MGM 247) [2]

Murdoch is correct that, similar to Zen, Weil aimed at an ultimate removal of duality. But she does not remark on the important difference between Weil and Zen on this point,

possibly because what bothers her is the shared removal of the self, and not whether the self is unified or just forgotten like a dream. In Weil, the self needs to disappear in order to dissolve itself into reality, or God: 'I am all. But this particular "I" is God. And it is not an "I". Evil makes distinctions, it prevents God from being equivalent to all' (GG 31) [25]. Thus, as Eric O. Springsted [36] has noted, we can think of Weil not as a dualist as she is often represented, but as holding to a 'monistic mystery', where 'mystery here stands as the locus of the harmony of the world' (10).

When we perceive the world attentively, there is no question of an 'I'. This can be the best description of successful attention and unselfing/decreation in our experience. However, precisely for its refusal to assign or remove any substance from the self, the Zen way is ultimately not one that either Murdoch of Weil can embrace, for opposite reasons. For Murdoch, a sense of self is necessary for her very conception of morality: as the 'locus' of morality, as that which others love in us, as that which presents to us the world with varying degrees of clarity. For Weil, it is important that the self is recognised as illusory and, hence, as lacking reality, not even as a non-unitary flux of experience, but as that which we identify precisely in order to give up, where giving it up is part and parcel of the ethical end-point.[23] Unlike Zen, Weil's idea of attention needs a metaphysics, with two key aspects. One is the unity, or monism, where the individual self is absorbed into the reality of God, or the real; while Zen, making no positive pronouncements in denying separation, does not need to claim unity. The other is the idea that the self, although illusory, needs to be given up: this renunciation, necessary for the ethical in Weil, is absent in Zen, where the illusory nature of self, once discovered, leaves us with the experience that there is nothing to renounce.

## 6. Conclusions

The related concepts of moral perception, attention, and the self are doubtlessly maverick in both Murdoch and Weil. That is part of their interest. On the one hand, they assign fundamental moral value to the activity of consciousness, which takes on intrinsic moral value and makes the value of action dependent on it; the field of morality is nothing short of our every thought and perception. On the other, they do not doubt that value is real and as much a part of the tissue of reality as shapes and colours. This claim is so fundamental to their philosophies that they do not feel the need to offer the kinds of arguments that would engage properly with contemporary philosophy, thus, making them hard to place. However, the moral task they both suggest is as valid as ever: to see the world as it is and, at the same time, to love the world, for perception is not possible without love. This love is attention, and it is also the withdrawal of the self to make space for reality and tend-toward the other. Nonetheless, as I have been arguing, Murdoch's and Weil's visions on this topic differ in important ways, starting from the role of the self. This difference has significant implications, I have suggested, for the meaning of attention, moral perception, and the general nature of the moral task, so we should not ignore them. In concluding, I will draw out some of these implications.

In both philosophers, attention is what allows us to 'join the world' (cf. MGM 496) [2]. Concomitantly, the self, which is what separates us, needs to retreat. This comes with an experience of clear perception as well as joy in 'losing oneself' in the object (cf. [37]). However, the meaning of this 'union' is very different depending on whether the self is real, historical, and with some continuity, or an illusion that by its very being creates the separation. The metaphysics here determines the experience and meaning of joining. In Murdoch, it is akin to a joining of two individuals, who put aside their differences to fully enjoy the sense of immediacy and togetherness, or the sense of presence in the perception of a reality that presents itself to us immediately, undistorted, in flow. In Weil, it is a dissolution of one into the other, of the individual into reality, so that what is real swallows the person and her point of view, and *that* is the joy.

The moral task of attention, further, draws on two different conceptions of the self: on the one hand, we will think of it as an improvement of an individual; on the other, as the self-withdrawal of an illusory self. That gives us the picture of *moral progress* that Murdoch offers, and of radical *moral transformation* found in Weil. Murdoch, granted, does make room for transformation and 'periagoge', but that is the transformation of the individual, and much of her focus is on the path, the small piecemeal activity of tiny improvements, with an important cumulative effect. For Weil, we are either pulled by grace or by gravity. Although reaching goodness is in fact impossible, anything short of that is failure. Attention does not aim at self-improvement but at self-effacement, and in true perception we are not partly withdrawn, but absent.[24] To return to Zen, it is appropriate that Murdoch picks up on Weil's remarks on the *koan*: a paradox that is only grasped when giving up conceptual thought and the self's effort, and is not grasped gradually, but occasions a sudden transformation. Nonetheless, I have suggested that the similarities between Weil and Zen are limited, although both are useful, self-less contrasts to Murdoch's ideas of attention and unselfing. In contrast to both Murdoch and Weil, on the other hand, Zen Buddhism offers a view of attention that is phenomenologically attractive, capturing the immediacy and fullness of presence that Murdoch and Weil describe, and without a metaphysical commitment that creates difficulties for both 'tame' and 'radical' views. This lack of metaphysics can be a strength but also a weakness, for it is metaphysics which, in both of the other views, explains moral progress on the one hand, and the imperative to love reality on the other.

Moral perception, finally, has a different nature depending on which view we follow. From the Murdochian view, perception is an *interaction* of self and world, one that, thanks to the use of the imagination and virtues, can be truthful. All perception for her has this structure, so that there is no difference between moral and non-moral perception. From the Weilian view, moral perception is also ubiquitous, but for the opposite reason: what makes it moral is the value of reality itself, which is disclosed not through our morally alive consciousness, but precisely through its absence. Perception is moral insofar as the value of reality shines through without my interference. 'May I disappear in order that those things that I see may become perfect in their beauty from the very fact that they are no longer things that I see' (GG 42) [25]. This may strike us, once again, as both utterly consistent and impossible, but for Weil, contradiction and impossibility are the marks of truth. *All* of reality has value in its very existence, but we have to learn to see it. To see it is to give up ourselves, down to our very capacity to see. This is, indeed, radical moral perception.

**Funding:** This research has received funding from the European Union's Horizon 2020 research and innovation programme under the Marie Skłodowska-Curie grant agreement No. 101026701.

**Conflicts of Interest:** The author declares no conflict of interest.

## Notes

1　　Henceforth IP.

2　　Henceforth MGM.

3　　Henceforth VCM.

4　　See for instance 'The Darkness of Practical Reason': 'A constructive activity of imagination and attention "introduces" value into the world which we confront. We have already partly willed our world when we come to look at it; and we must admit moral responsibility for this "fabricated" world' (DPR 201) [38].

5　　Where for Murdoch, Platonically, truth is good, and truth-seeking is good-making.

6　　Which she attributes, among others, to Wright, Cullison, Audi, and Cowan [39–42].

7　　See Holland, Denham, Chappell [43–45]. A recent account of moral perception that both caters to the importance of the moral sensibility of the individual and takes attention to be central is 'attentional moral perception', proposed by Vance and Werner [46], whereby attention is sensitive to moral 'difference makers' and moral thinking is sensitive to our attentional patterns. Although this account includes elements that Murdoch would accept, the underlying concept of attention—namely, the familiar concept of attention as selectivity—constitutes only a small part of Murdochian attention. Murdoch would agree that what we attend to matters, partly because salience leads to a different understanding of the situation. But this is only part of the story. For Murdoch,

not just the objects of perception, but the quality of perception, is what is morally at stake. In attention, we do not only select one object as opposed to another, but attention itself determines *what* we see. That is because, for Murdoch, attention is not just what enables perception, but a special mode of perception.

8    Henceforth OGG.

9    In some instances, Murdoch talks about the 'ego' in this context rather than the self, but in many cases she attributes the same qualities to the 'self', making a clear-cut distinction between self and ego, in Murdoch's exegesis, difficult to maintain thoughout.

10    These reflections can be found throughout her work, but see e.g., these remarks: 'the central concept of morality is "the individual"' (IP 323) [1] and 'the (daily, hourly, minutely) attempted purification of consciousness [is] the central and fundamental "arena" of morality' (MGM 193) [2].

11    See [8] for a defence of the role of experience and knowledge in perception see Dancy [47] and, in Murdoch specifically, Mole [15].

12    In psychology, the selectivity model was first introduced by Broadbent [48], whose 'bottleneck' model then faced severe criticism, but the selectivity model was retained e.g., by Eysenck and Keane [49], Treisman [50].

13    As Murdoch writes of M: 'M's activity is peculiarly *her own. Its details are the details of *this* personality' (IP 317) [1].

14    This paper is a development of the distinction I present in Chapters 2 and 3 of [19], with the aim of presenting a closer comparison of Murdoch's and Weil's models of attention and moral perception, which in the previous work are presented separately.

15    The idea of attention is, however, partly lost in the English translations, where the polysemy of the French gives way to choices such as *Waiting for God, Waiting on God*, but also, more closely perhaps, *Awaiting God.*

16    Henceforth GG.

17    According to Weil (inheriting the Kabbalistic idea of Tzimtzum) God created the world through self-withdrawal, which is also a supreme act of love. (In decreation we imitate this act.) Hence the world carries God's presence through God's absence. On presence-as-absence in attention see Pirruccello [51].

18    See Vetö [52] (Chapter 2) for an exhaustive description of Weil's metaphysics of attention.

19    The treatment of these extremely important themes is here necessarily cursory. See Sook Cha's [28] book for a helpful reading.

20    See e.g., MGM 25 [2] and GG 118 [25].

21    It should be clear that it is the metaphysical, not the physical, self that Weil intends to dismantle. Despite some mostly biographical speculations, she did not advocate suicide but rather the opposite. See e.g., GG 52: 'Suicide is probably never anything else [than imaginary], and that is why it is forbidden' [25].

22    Murdoch's ultimate distancing from Zen has been acknowledged by commentators, such as Robjant [53] (p. 1005) and Beran and Marchal [32] (p. 183). The distancing of Murdoch from Weil was already apparent in her earlier discussions of Weil, her radio talk from 1951 [54] and her review of Weil's Notebooks from 1956 (KV) [55]. In the radio talk Murdoch says: 'One simply cannot say that the realm of imagination is the realm of delusion. To cut through human nature so harshly is to leave the pure portion of it so unrecognizably inhuman that in the end we have learnt nothing' [54] (p. 15). The central point of separation, it seems, remains the same, and has to do with the self.

23    This is also why Weil felt greater and declared affinity with Hinduism, with its metaphysics where the true self, *ātman*, is the universal Self, hence the individual self is illusory, but the universal self has a reality in which we participate losing individuality. Vivienne Blackburn [56] has found evidence that Weil's very concept of attention was elaborated with the contribution of her study of Hinduism (p. 261) and that her first notes about Hinduism concern the *ātman.*

24    Murdoch remarks on her distance from Weil explicitly in her radio talk on Weil, where she makes it clear that while Weil cares about the ideal, she also wants to keep the path to the ideal: 'Her [Weil's] picture of the intellect as waiting upon truth in order to *accept* it is in a way exact. But intellectual work is not only attention—it is also setting the stage for attention. And where most human matters are concerned we are never able to finish for long with the task of setting the stage' [54] (p. 15).

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
