# Peer review of "Perception, Self, and Zen: On Iris Murdoch and the Taming of Simone Weil"

_philosophies, doi:10.3390/philosophies8040064_

Round 1

Reviewer 1 Report

This is a very interesting, well written, insightful paper, both as an independent reflection on the issues surrounding the topic of moral perception and as a contribution to the recently growing Iris Murdoch scholarship. There are a few typos, which I'm sure a more careful reading will reveal. My only more substantive comment is this: it is not perfectly clear what role the discussion of the Zen Budhist conception of attention is meant to play in the paper. At times it seems that it is there to highlight the difference between the Murdochian and the Weilian conceptions of attention and at other times it seems that it is there to show us the best way to think about the matter. In any case, I think that the paper offers valuable service both to the discussions of moral perception and to the literature on Murdoch. I enjoyed reading it and learned a lot from it. 

Author Response

I would like to thank the reviewer for their helpful feedback. As for the more substantive comment, I have added a section that aims to clarifies the role of Zen. Zen is introduced for two main reasons: one is that Murdoch herself invokes Zen to discuss her refusal to give up the self (lines 385-89; 414-17); the other is that Zen provides an alternative to both Murdoch and Weil, while sharing more similarities to Weil (499-505), an alternative that is both attractive and problematic in difference ways (550-562).

Author Response

I would like to thank the reviewer for their helpful feedback. I have corrected the typos, the references, and the grammatical errors they pointed out. I am grateful for the careful reading of the paper and the opportunity to make these changes.

Concerning the more substantive points, I have made the following changes (and additions in lines 201-218):

1 The lack of evidence for assimilating Weil’s attention into Murdoch’s: I have made two changes to address this point. First, I have toned down the ‘accusation’, offering what is indeed a better objection, namely that many authors unproblematically mention Murdoch’s attention as inspired by Weil, but do not say that there are important differences. Second, I have added several references for this (lines 205-6).

2 The unfair reading of Broackes: it is true that Broackes mentions an important difference concerning the individual. I have quoted that sentence, and specified that although he acknowledges that difference, he still sees Murdoch’s project as fundamentally similar to Weil. This has allowed me to clarify the nature of my objection and make it more specific. I have also added a reference to Caprioglio Panizza 2022 and stated that the distinction, and the labels, come from there.

3 Concerning the difference Zen/Weil: I have added a section to explain how the Zen overcoming of dualism is different from Weil’s. (499-505 and 550-564).

4 I have added some sentences to explain why Murdoch did not comment on the difference between Weil and Zen in the passages analysed. (481-84)

Concerning the specific comments:

  1. Lines 69ff: it seems tangential to the reconstruction to maintain that Murdoch’s notion of perception does not fit into contemporary discussions. This point should, perhaps, either be eliminated or expanded and more fully integrated it into the discussion.
  • I agree that this point is not essential to the distinction the paper aims to make; however, I am hesitant to remove it because, in the context of the special issue, it helps to place the kind of moral perception being discussed in the context of the more familiar kinds that are addressed in the SI.
  1. Lines 285-294: this also seems a tangential point.
  • I see the worry, so I have shortened this point and removed the footnote, bringing part of it into the text. I like this point, but I accept that it’s not essential, so if the reviewer feels strongly that it should go, I can remove it.

Reviewer 3 Report

This is a well-written, interesting, and insightful paper on Murdoch, Weil, and Buddhism that demonstrates a high level of engagement with both the primary texts and the recent scholarly literature. I found it very interesting and illuminating to read, and I believe that it would make an important and valuable contribution to the literature. For that reason, I strongly recommend publication. 

I have only one significant suggestion for improvement as far as the content of the paper is concerned, and a few small suggested edits (mostly arising from typos). 

The main area in which I think the argument could be improved is in the author's treatment of metaphor and metaphorical language. On p 2 the author writes that "it is possible and tempting to take Murdoch's frequent use of visual terms only as metaphors" and on p 3 they write that in speaking of the self as an obstacle to attention Murdoch "is not only talking metaphorically". This suggests that if we thought that Murdoch were "only talking metaphorically" we would have reason to take her claims less seriously, but such a reading would be completely at odds with what Murdoch herself says about metaphor and metaphorical language. (See for example "The Sovereignty of Good Over Other Concepts", where Murdoch writes that "metaphors are not merely peripheral decorations or even useful models, they are fundamental forms of our awareness of our condition" (363), and describes her own project in terms of providing "explanatory and persuasive metaphors" (364).) I suggest that the author revise their language here to bring it closer into line with Murdoch's own views on the status of metaphor (which is something that I think can be done without any substantial changes to the main line of argument of the paper).

A few small edits to typos:

p 6 line 234: "desires" should be "desire"

p 7 line 278: "these action" should be "these actions"

p8 line 335: "calling oneself" should be "calling herself"

I would also suggest that the author consider rewriting the sentence that begins on p 6 line 221 , as the wording ("Meaningfully, one of the main collections of Weil's writings...") seemed awkward to me. 

Author Response

I would like to thank the reviewer for helpful suggestions and comments.

I have revised the paper and kept my eyes wide open for typos, which hopefully are now gone.

As for the more substantive comment, I agree entirely that metaphors are not mere additions for Murdoch, and that in the attempt to describe vision to an audience not familiar with Murdoch, my comment has awkwardly dismissed the role of metaphor. I have therefore deleted the comment at the start of section 2 and added a clarification on lines 61-62 (borrowing, if I may, the useful reference supplied in the review).
